# An Optimized Stacking Ensemble Model for Phishing Websites Detection

Mohammed Al-Sarem [1], Faisal Saeed [1,*], Zeyad Ghaleb Al-Mekhlafi [2,*], Badiea Abdulkarem Mohammed [2], Tawfik Al-Hadhrami [3,*], Mohammad T. Alshammari [2], Abdulrahman Alreshidi [2] and Talal Sarheed Alshammari [2]

1   College of Computer Science and Engineering, Taibah University, Medina 42353, Saudi Arabia; msarem@taibahu.edu.sa
2   College of Computer Science and Engineering, University of Ha'il, Ha'il 81481, Saudi Arabia; b.alshaibani@uoh.edu.sa (B.A.M.); md.alshammari@uoh.edu.sa (M.T.A.); ab.alreshidi@uoh.edu.sa (A.A.); talal.alshammari@uoh.edu.sa (T.S.A.)
3   School of Science and Technology, Nottingham Trent University, Nottingham NG11 8 NS, UK
*   Correspondence: fsaeed@taibahu.edu.sa (F.S.); ziadgh2003@hotmail.com (Z.G.A.-M.); tawfik.al-hadhrami@ntu.ac.uk (T.A.-H.)

**Abstract:** Security attacks on legitimate websites to steal users' information, known as phishing attacks, have been increasing. This kind of attack does not just affect individuals' or organisations' websites. Although several detection methods for phishing websites have been proposed using machine learning, deep learning, and other approaches, their detection accuracy still needs to be enhanced. This paper proposes an optimized stacking ensemble method for phishing website detection. The optimisation was carried out using a genetic algorithm (GA) to tune the parameters of several ensemble machine learning methods, including random forests, AdaBoost, XGBoost, Bagging, GradientBoost, and LightGBM. The optimized classifiers were then ranked, and the best three models were chosen as base classifiers of a stacking ensemble method. The experiments were conducted on three phishing website datasets that consisted of both phishing websites and legitimate websites—the Phishing Websites Data Set from UCI (Dataset 1); Phishing Dataset for Machine Learning from Mendeley (Dataset 2, and Datasets for Phishing Websites Detection from Mendeley (Dataset 3). The experimental results showed an improvement using the optimized stacking ensemble method, where the detection accuracy reached 97.16%, 98.58%, and 97.39% for Dataset 1, Dataset 2, and Dataset 3, respectively.

**Keywords:** ensemble classifiers; phishing websites; genetic algorithm; optimization methods





## 1. Introduction

One of the most dangerous cybercrimes is phishing, where the user's information and credentials are stolen using fake emails or websites that are sent to the target and look like legitimate ones. Phishing attacks have been increasing over the years, and affect many internet users. In this type of attack, the phisher selects any organisation as a target, and then develops a phishing website that is similar to the organisation's legitimate website. The phisher then sends several spam emails or posts these links using social media or any communication medium to many internet users, who may click on these links and be redirected to the phishing website [1].

Phishing is one type of social engineering attack that targets many organisations' websites on the internet. It can also attack internet of things (IoT) environments, in which the devices are highly interconnected, and these threats can affect organizations' privacy and data. IoT sensors are considered to be an easy medium for attackers. According to [2], attackers sent several spam emails, and it was found that refrigerators, televisions, and routers were among the 25% of devices that hosted them. In addition, hackers in the IoT

environment may not need to send a virus or Trojan, as they can use the software in the thingbots for spreading spam emails without the user knowing, as this may not affect the functionality of IoT devices [3]. Many methods have been introduced to make the IoT environment more secure, but there is currently no effective method for detecting phishing emails [1,4]. Several studies have been conducted in order to propose approaches and methods for detecting phishing websites for the IoT environment. For instance, Wei et al. [5] introduced a lightweight deep learning method in order to provide a phishing detection sensor that could work in real time with energy-saving features. If using this proposed system, there is no need to install anti-phishing software on every IoT device. However, the designed sensor is only needed for one location (such as an office) between the devices and the router. In addition, this model can be directly installed on the router because of its high efficiency.

Deep learning methods have been widely investigated for detecting phishing websites. For instance, Somesha et al. [6] applied several models for phishing detection, which included convolution neural network (CNN), deep neural network (DNN), and long short-term memory (LSTM) models. The applied models obtained a good detection rate, with an accuracy of 99.57% for LSTM. These models used only one third-party service feature, in order to make the model robust and efficient. In another study, Ali and Ahmed [7] introduced a hybrid intelligence method for predicting phishing websites, in which a genetic algorithm (GA) was utilized to identify the optimal weights for website features and select the most important ones. These features were used to train deep neural networks to predict the phishing URLs. The results showed that the proposed approach obtained significant improvements in terms of accuracy, specificity, sensitivity, and other metrics compared to other state-of-the-art methods.

In a different approach, several machine learning methods were used to detect the phishing websites. For instance, Chiew et al. [8] introduced a framework based on feature selection and machine learning methods for detecting phishing, named hybrid ensemble feature selection. In this method, the primary feature subsets were obtained using the cumulative distribution function gradient, and these subsets were used to obtain the secondary feature subsets using a data perturbation ensemble. The proposed model used only 20.8% of the original features, and obtained an accuracy of 94.6% using the Random Forests method. Similarly, Rao and Pais [9] introduced an efficient model based on feature selection and machine learning; in order to improve the limitations of the currently used phishing detection methods, they obtained the heuristic features from the websites' URLs, source codes, and third-party services. Eight machine learning methods were used to evaluate the proposed model, and Random Forests obtained the best accuracy (99.31%). In addition, Ali and Malebary [10] proposed a novel phishing detection model by utilizing the particle swarm optimization method in order to weight the websites' features, which helped to identify the importance of their contributions towards differentiating the phishing websites from legitimate ones. The results showed that this model led to outstanding enhancements in terms of accuracy and other performance metrics for several machine learning methods.

This paper proposes a model which is known as an optimized ensemble classification model for detecting phishing websites. A genetic algorithm (GA) is used to optimize the performance of several ensemble classifiers. Then, the best optimized classifiers are used as base classifiers for the stacking ensemble method. The method includes three main phases: training, ranking, and testing. In the training phase, random forests, AdaBoost, XGBoost, Bagging, GradientBoost, and LightGBM are trained without applying an optimization method. These classifiers are then optimized using the genetic algorithm, which selects the optimal values of parameters for several ensemble models. The optimized classifiers are then ranked and used as base classifiers for the stacking ensemble method. Finally, new websites are collected and used as a testing dataset in order to predict the final class label of these websites.

The rest of this paper is organized as follows: Section 2 gives an overview of the related work. Section 3 provides details about the materials and methods. Section 4 presents the experimental results, which are analysed, discussed, and compared with related works. The paper concludes with a summary of the outcomes of the proposed method and suggestions for future work.

## 2. Related Works

### 2.1. Recognizing Phishing Attacks in the IoT

There are serious issues regarding the security of the IoT web, as there are billions of devices (network objects and sensors) that are connected to the internet [11]. Thus, there is a strong need to protect these IoT data from various types of attacks, including phishing. Gupta et al. [1] illustrated how advanced infrastructures such as the internet of things (IoT) are considered a target for phishing attacks. Tsiknas et al. [12] reviewed the main cyber threats to the industrial internet of things (IIoT), and found that they originate from five types of attacks: phishing, ransomware, system attacks, supply chain, and protocol. According to Tsiknas et al. [12], for critical infrastructure such as the IoT, phishers apply compromised attacks—an advanced method that combines social engineering and includes zero-day malware and other features that are designed on remote websites and then attack IIoT systems. The malicious attacker uses the front-end level for accessing the IIoT.

Several methods have been proposed to detect phishing websites in the IoT environment. Parra et al. [13] proposed a cloud- and deep-learning-based framework that includes two mechanisms: a distributed convolutional neural network, and cloud-based temporal long short-term memory. The first mechanism was used for detecting phishing as an IoT microsecurity device, while the second mechanism was used on the back end to detect notnet attacks and ingest CNN embeddings for detecting distributed phishing attacks on several IoT devices. The experimental results showed that the first mechanism could obtain a detection accuracy of 94.3% running the CNN model, and an F-1 score of 93.58% for phishing attacks.

Mao et al. [14] discussed the main security issues in smart internet of things (IoT) systems, and found that phishing is one of the most common types of attacks. In order to detect these phishing websites, they developed an automated page-layout-based method that includes machine learning methods. The method is based on aggregation analysis for obtaining the page layout similarity, which helps in detecting phishing websites. Four ML methods were applied in these experiments, and the results obtained showed enhanced accuracy.

The security issues in the IoT were discussed in detail by Virat et al. [15], who argued that the main challenge with IoT security is that its devices are not intelligent, which makes the task of solving these issues difficult, requiring appropriate detection methods. In addition, Deogirikar and Vidhate [16] surveyed various vulnerabilities that put the IoT as a technology in danger. They reviewed various IoT attacks and discussed their efficiency and damage level in the IoT, and concluded that extensive research is required in order to come up with effective solutions.

In addition, deep learning methods were also investigated for protecting internet of things (IoT) devices against several attacks, such as distributed denial-of-service (DDoS), phishing, and spamming campaigns. In [17], a stacked deep learning method was introduced to detect malicious traffic attacks affecting IoT devices. This proposed method showed a good ability to detect benign and malicious traffic data, and obtained a higher detection rate in real time compared with other methods.

### 2.2. Machine-Learning-Based Detection Methods

Artificial intelligence (AI) and machine learning (ML) have been widely used as detection methods for several cyber security issues. For phishing website detection, several AI- and ML-based methods with good detection performance have been proposed. For instance, Alsariera et al. [18] proposed new schemes based on AI that considered new

methods for the mitigation of phishing. They introduced four meta-learning techniques based on the extra-tree-based classifier and applied them to phishing website datasets. The experimental results showed that the proposed models obtained an accuracy of 97%, and reduced the false positive rate to 0.028.

Jain and Gupta [19] proposed a new method for detecting phishing websites based on the hyperlinks located in the websites' HTML code. This method combines several novel features of hyperlinks, and divides them into 12 types for training ML models. This method was applied to a phishing website dataset using several ML classifiers. The experimental results showed that the proposed model obtained 98.4% accuracy using a logistic regression classifier. This method is a client-side solution, which does not require any third-party support. Feng [20] introduced a new a model for phishing website detection using a neural network. The Monte Carlo technique was used in the training phase, and in the testing phase the accuracy reached 97.71% while the false positive rate reached 1.7%, indicating that the proposed model is capable of detecting phishing websites effectively compared to other machine learning methods.

Aburub and Hadi [21] used association rules to detect phishing websites. They used a dataset containing 10,068 instances of legitimate and phishing websites, and applied the phishing multi-class association rule method, which was compared to other associative classification methods. The experimental results showed that the proposed methods obtained an acceptable detection rate. Similarly, other ML-based methods have been applied utilizing feature selection methods [22,23], ensemble classifiers [24], hybrid methods of deep learning and machine learning [25], and other methods.

As can be shown from the previous studies on detecting phishing websites, the effectiveness of the detection still needs to be enhanced. For instance, Azeez et al. [26] mentioned that the current applied methods to handle phishing websites are not sufficient. Thus, they introduced the PhishDetect method, which identifies phishing attacks by using URL consistency features. This proposed method checks the PhishTank database in order to verify whether the URL exists, then considers it to be a phishing website if not. This method requires updating the PhishTank database frequently. In addition, Azeez et al. [27] proposed a system for detecting malicious URLs on Twitter. This study examined the correlation of URL redirect chains obtained from Twitter, and then a naive Bayes classifier was used on these data, with an accuracy of 90%. An interesting comparative study was conducted by Osho et al. [28] to investigate the performance of several machine learning methods for the detection of phishing websites. They found that the random forests method outperforms the existing methods, and achieves an accuracy of 97.3%.

However, some proposed methods were applied to small- or medium-sized datasets, while other proposed methods were applied to only one dataset (websites or emails). Therefore, there is a need to conduct further analysis on detecting phishing websites using more datasets with many benign and malicious websites.

## 3. Materials and Methods

In this section, the proposed genetic-algorithm-based ensemble classifier approach for improving phishing website detection is presented and explained. Figure 1 presents the methodology that we followed in this study. The methodology consists of three main phases: the training, ranking, and testing phases. In the training phase, random forests, AdaBoost, XGBoost, Bagging, GradientBoost, and LightGBM were trained without optimization. The reason behind this is twofold: on the one hand, to obtain a general insight into the performance of ensemble classifiers before optimizing them, and on the other hand, to explore which of the phishing websites' characteristics is most useful. The aforementioned classifiers were then optimized using the genetic algorithm. Here, the genetic algorithm was used for selecting the optimal values of model parameters in order to improve the overall accuracy of the proposed model. Later, in the ranking phase, the optimized classifiers were ranked and used as a base classifier for the ensemble classifier—the stacking method. In the testing phase, new websites were collected and used as testing

data. Figure 1 refers to this phase as the detection phase, as these steps will be applied to any website in future in order to detect whether it is a benign or malicious website. In order to extract the features of the websites, we followed the methodology presented in [29]. A set of benign and malicious websites was collected from the malware and phishing blacklist of the PhishTank database of verified phishing pages [30]. In order to extract the same features as those used in the training dataset (HTML- and JavaScript-based features, address-bar-based features, domain-based features, and abnormality-based features), a Python script was written using the Beautiful Soup, ipaddress, urllib, request, and Whois libraries. Later, all of these features were fed into the classifiers in order to predict the final class label of the website.

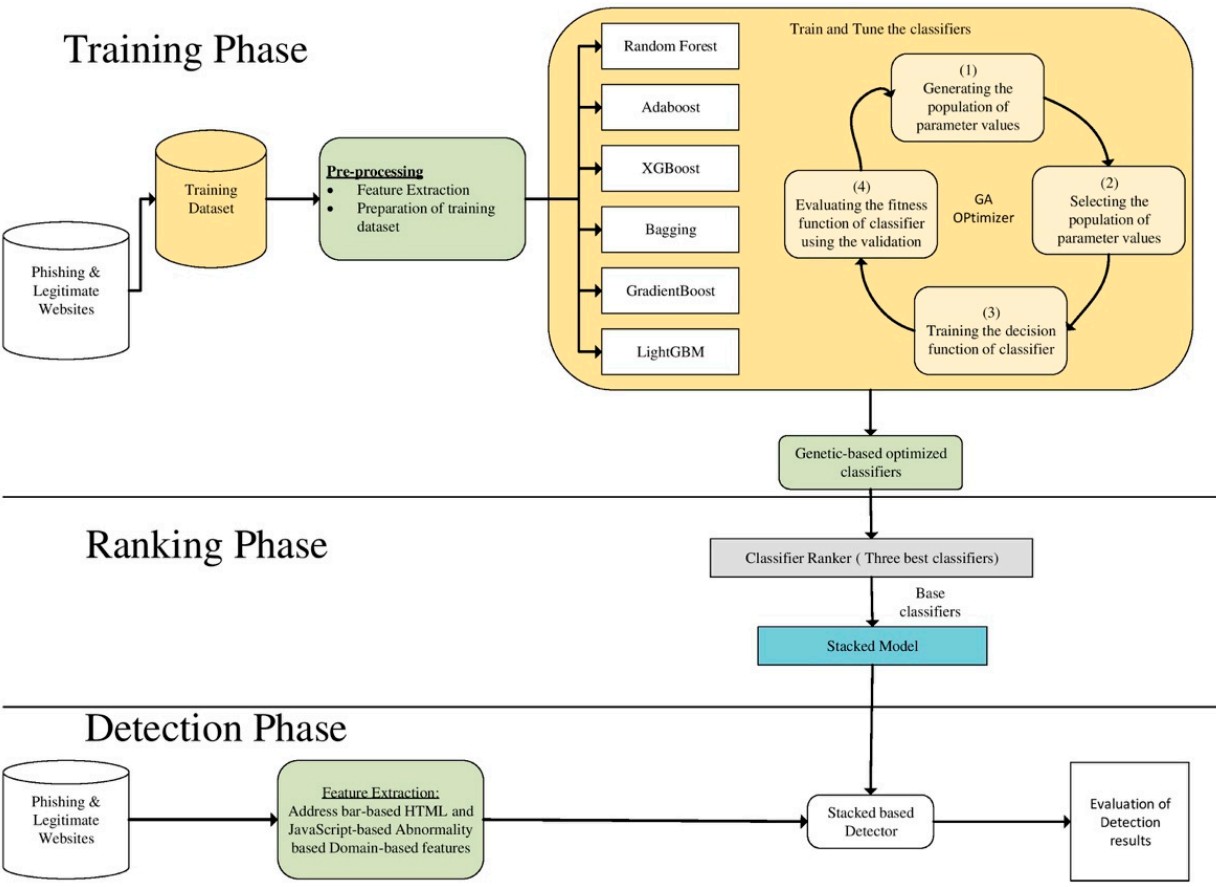

**Figure 1.** The proposed optimized stacking ensemble model for phishing website detection.

*The Dataset and Experimental Design*

The experimental part of this work was conducted on three publicly available datasets—the Phishing Websites Data Set from UCI (Dataset 1) [31], the Phishing Dataset for Machine Learning from Mendeley (Dataset 2) [32], and Datasets for Phishing Websites Detection from Mendeley (Dataset 3) [33]. To conduct the experiment, the script was written in Python 3.6 using an Anaconda environment on the 64-bit Windows 10 operating system. Dataset 1 consists of 44% phishing websites (4898) and 56% legitimate websites (6157). Since the dataset is quite imbalanced, the oversampling technique was used to increase the size of the minority class. The dataset contains 30 features, which can be categorized into four groups: (1) 12 address-bar-based features, (2) 5 HTML- and JavaScript-based features, (3) 6 abnormality-based features, and (4) 7 domain-based features. Table 1 presents the names of these features and the Python library used for extracting each one in the testing phase.

**Table 1.** Feature description for phishing websites.

| Feature Category | Feature Name | Description | Python Library Used |
|---|---|---|---|
| Address-bar-based | having_IP_Address | Using the IP Address | IPaddress Urllib Re Datetime BeautifulSoup Socket |
| | URL_Length | Long URL to hide the suspicious part | |
| | Shortening_Service | Using shortening service | |
| | having_At_Symbol | URL having @ symbol | |
| | double_slash_redirecting | URL uses "//" symbol | |
| | Prefix_Suffix | Add prefix or suffix separated by (-) | |
| | having_Sub_Domain | Website has subdomain or multi-subdomain | |
| | SSLfinal_State | Age of SSL certificate | |
| | Domain_registeration_length | Domain registration length | |
| | Favicon | Associated graphic image (icon) with webpage | |
| | Port | Open port | |
| | HTTPS_token | Presence of HTTP/HTTPS in domain name | |
| HTML- and JavaScript-based | Redirect | How many times a website has been redirected | Request BeautifulSoup |
| | on_mouseover | Effect of mouse over on status bar | |
| | RightClick | Disabling right click | |
| | popUpWindow | Using pop-up window to submit personal information | |
| | Iframe | Using Iframe | |
| Abnormality based | Request_URL | % of external objects contained within a webpage | BeautifulSoup Re WHOIS |
| | URL_of_Anchor | % of URL Anchor (<a> tag) | |
| | Links_in_tags | % of links in <meta>, <script> and <link> | |
| | SFH | Server from Handler | |
| | Submitting_to_email | Submit user information using mail or mailto | |
| Domain-based features | Abnormal_URL | Host name in URL | WHOIS Urllib BeautifulSoup |
| | age_of_domain | Age of the website | |
| | DNSRecord | Website in WHOIS dataset | |
| | web_traffic | Popularity of the website | |
| | Page_Rank | Page Rank | |
| | Google_Index | Google Index | |
| | Links_pointing_to_page | # of links pointing to page | |
| | Statistical_report' | found in statistical reports | |
| | Result | Website is classified as phishing or legitimate | |

In addition, Dataset 2 includes 48 features extracted from 5000 phishing websites and 5000 legitimate websites, while Dataset 3 includes 111 features extracted from 30,647 phishing websites and 58,000 legitimate websites. More descriptions about these datasets can be obtained from [32,33].

In order to evaluate the performance of the proposed ensemble model, the following performance measures were used: classification accuracy, precision, recall (the detection rate), F1 score, false positive rate (FPR), and false negative rate (FNR). These measures are commonly used by researchers to evaluate the performance of phishing website detection systems [10]. In order to precisely assess the proposed method, all of the conducted experiments including optimized and non-optimized classifiers were validated using 10-fold cross-validation. The results of each fold were also normalized. $P = (95.37/(95.37 + 1.2)$, $R = 95.37/(95.37 + 4.63)$.

## 4. Results and Discussion

This section describes the experimental results for each technique, before presenting and discussing comparisons with the related works.

### 4.1. Experimental Results of the Ensemble Classifiers without Optimization

As mentioned earlier, a set of ensemble classifiers was trained using 10-fold cross-validation. We first conducted the experiment without involving the optimization using the GA. The performance of the classifiers with default configurations is presented in Tables 2–4 for Dataset 1, Dataset 2, and Dataset 3, respectively. For Dataset 1, the random forests classifier yielded the best performance compared with the other classifiers in terms of accuracy, precision, recall, and F-score; it achieved 97.02% accuracy. The Bagging classifier also achieved good accuracy, with 96.73%, followed by the LightGBM classifier, with accuracy of 96.53%. The remaining classifiers obtained accuracy between 93% and 94.61%. Meanwhile, in Dataset 2, the LightGBM classifier obtained the best performance compared to other classifiers in terms of accuracy, precision, recall and F-score. The random forests classifier obtained the second best performance using all evaluation measures for this dataset. Similarly to Dataset 1, the performance of Random Forests obtained the best results for Dataset 3 in terms of accuracy, recall, and F-score, as shown in Table 4.

**Table 2.** Performance of ensemble classifiers for Dataset 1.

| Measure | Random Forests (%) | AdaBoost (%) | XGBoost (%) | Bagging (%) | GradientBoost (%) | LightGBM (%) |
|---|---|---|---|---|---|---|
| Accuracy | **97.02** | 93.17 | 94.45 | 96.73 | 94.61 | 96.53 |
| Precision | **96.58** | 94.70 | 94.52 | 94.99 | 94.87 | 95.15 |
| Recall | **98.08** | 96.60 | 96.39 | 96.73 | 96.59 | 96.70 |
| F-Score | **97.49** | 95.71 | 95.50 | 95.90 | 95.76 | 95.95 |

**Table 3.** Performance of ensemble classifiers for Dataset 2.

| Measure | Random Forests (%) | AdaBoost (%) | XGBoost (%) | Bagging (%) | GradientBoost (%) | LightGBM (%) |
|---|---|---|---|---|---|---|
| Accuracy | 98.37 | 96.88 | 97.70 | 97.51 | 97.67 | **98.65** |
| Precision | 98.54 | 96.74 | 97.57 | 97.55 | 97.58 | **98.56** |
| Recall | 98.26 | 97.04 | 97.85 | 97.44 | 97.76 | **98.74** |
| F-Score | 98.39 | 96.88 | 97.71 | 97.46 | 97.67 | **98.65** |

**Table 4.** Performance of ensemble classifiers for Dataset 3.

| Measure | Random Forests (%) | AdaBoost (%) | XGBoost (%) | Bagging (%) | GradientBoost (%) | LightGBM (%) |
|---|---|---|---|---|---|---|
| Accuracy | **97.15** | 93.58 | 95.33 | 96.78 | 95.37 | 96.67 |
| Precision | 95.78 | 90.89 | 92.75 | **95.80** | 93.06 | 95.08 |
| Recall | **96.13** | 90.52 | 93.82 | 94.93 | 93.58 | 95.28 |
| F-Score | **95.90** | 90.70 | 93.28 | 95.33 | 93.32 | 95.18 |

Figure 2 shows the false positive rate (FPR) and false negative rate (FNP) for Dataset 1. It was notable that RF had the best FPR and FNP, with 0.05 and 0.02, respectively. The LightGBM classifier was the second best classifier in terms of FPR (0.068), followed by the GradientBoost classifier (0.07). In terms of FNR, the Random Forests classifier also yielded the lowest value (0.02), followed by AdaBoost and Bagging. Although the AdaBoost classifier had a lower FNR (0.032), its FPR values were higher than those of the LightGBM classifier, which means that there is a probability of raising a false alarm, in which a positive

result is given when the true value is negative. Similarly, as shown in Figures 3 and 4, the RF model obtained the best FPR and FNP for Dataset 2 and Dataset 3.

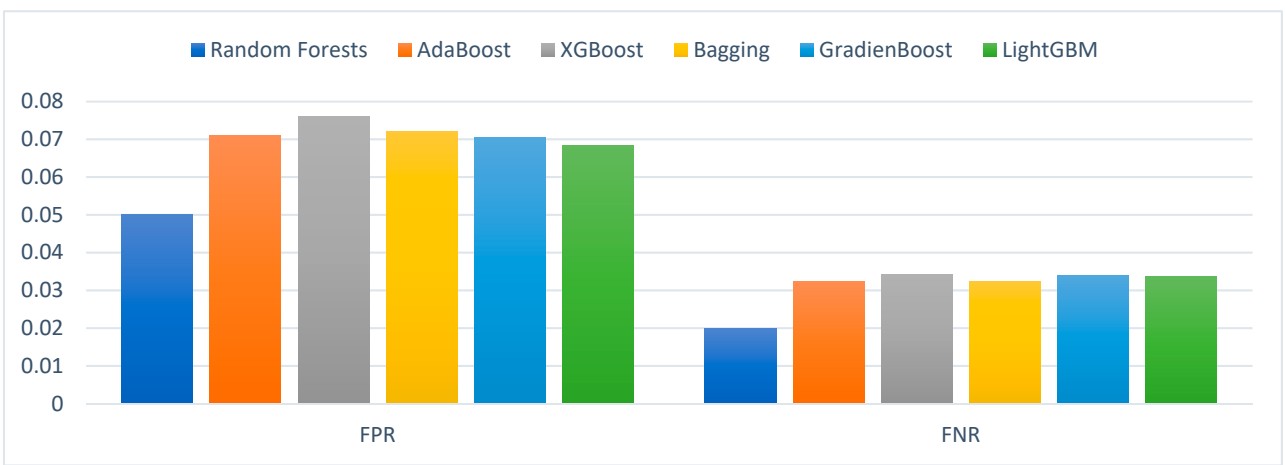

**Figure 2.** Comparisons of the FPR and FNR of ensemble methods for Dataset 1.

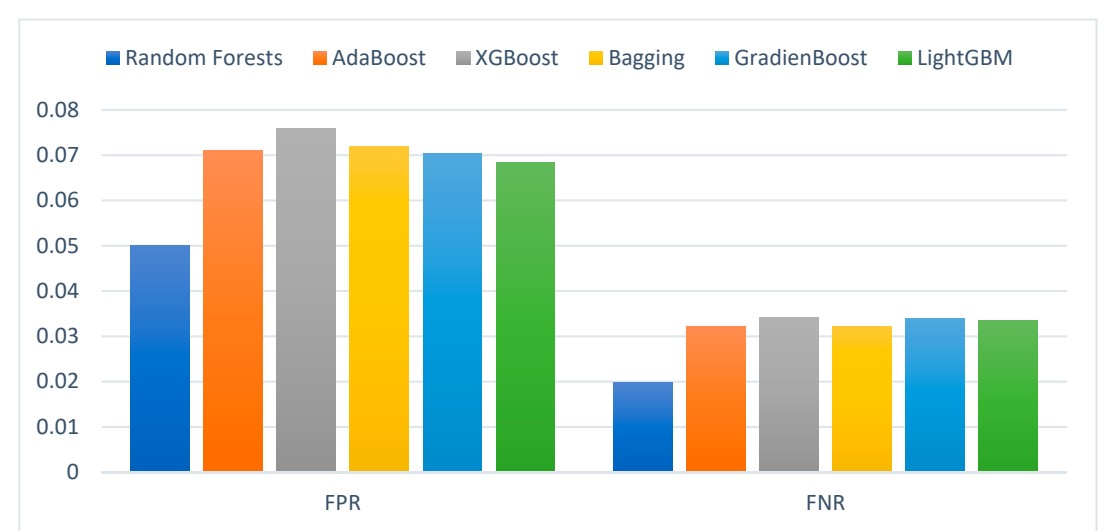

**Figure 3.** Comparisons of the FPR and FNR of ensemble methods for Dataset 2.

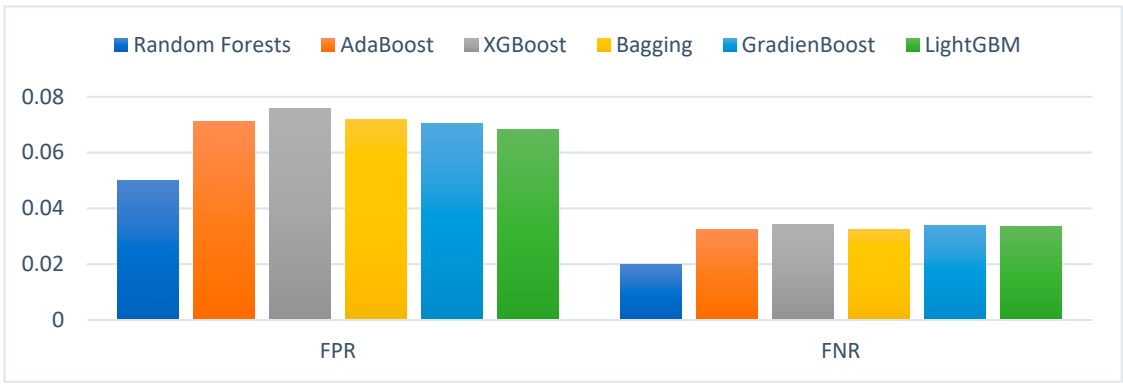

**Figure 4.** Comparisons of the FPR and FNR of ensemble methods for Dataset 3.

*4.2. Experimental Results of the GA-Based Ensemble Classifiers*

　　Although all of the classifiers showed good performance, there is still a need to adjust many of their parameters in order to achieve better evaluation scores. Adjusting such parameters for each classifier is relatively cumbersome. In this study, a genetic algorithm was used for tuning the classifiers' parameters. Gas have shown good results in the field of algorithm parameter searching [34]. We conducted the experiments using different parameters to configure the GA (which were used in our previous works and other studies), and the best ones were used in this study, as shown in Table 5.

**Table 5.** Parameter settings of the GA used in this paper.

| Parameter | Value |
|---|---|
| Generations | 10 |
| Population size | 24 |
| Mutation rate | 0.02 |
| Crossover rate | 0.5 |
| Early stop | 12 |

　　Since there are many parameters to adjust, Table 6 shows the list of adjusted parameters of each classifier and the optimized parameters found by the GA. Among all of the parameters, finding the optimal number of estimators and learning rate are the most critical parameters, which impact most highly on the performance of the classifier. XGBoost and GradientBoost gained a considerable improvement compared to the default parameters, as shown in Table 7. Meanwhile, the performance of both the LightGBM classifier and Random Forests was decreased.

**Table 6.** List of optimized parameters of the classifiers.

| Classifier Name | Adjusted Parameters | Best GA-Based Configuration |
|---|---|---|
| Random Forests | Criterion: ['entropy', 'gini']<br>max_depth: [10–1200] + [None]<br>max_features: ['auto', 'sqrt','log2', None]<br>min_samples_leaf: [4–12]<br>min_samples_split: [5–10]<br>n_estimators': [150–1200] | Criterion: entropy<br>max_depth: 142<br>min_samples_leaf: 4 min_samples_split: 5<br>n_estimators: 1200 |
| AdaBoost | n_estimators: [100–1200]<br>learning_rate: [$1 \times 10^{-3}, 1 \times 10^{-2}, 1 \times 10^{-1}, 0.5, 1.0$] | learning_rate: 0.1<br>n_estimators: 711 |
| XGBoost | n_estimators: [100–1200]<br>max_depth: [1–11],<br>learning_rate: [$1 \times 10^{-3}, 1 \times 10^{-2}, 1 \times 10^{-1}, 0.5, 1.$]<br>subsample: [0.05–1.01]<br>min_child_weight: [1–21] | learning_rate: 0.1<br>max_depth: 5<br>min_child_weight: 3.0<br>n_estimators: 588<br>subsample: 0.7 |
| Bagging | n_estimators: [100–1200]<br>max_samples: [0.1, 0.2, 0.3, 0.4, 0.5, 1.0, 1.1]<br>bootstrap: [True, False] | n_estimators: 1077<br>max_samples: 0.5<br>bootstrap: True |
| GradientBoost | n_estimators: [100–1200]<br>learning_rate: [$1 \times 10^{-3}, 1 \times 10^{-2}, 1 \times 10^{-1}, 0.5, 1.0$]<br>subsample: [0.05–1.01]<br>max_depth: [10–1200] + None<br>min_samples_split: [5–10]<br>min_samples_leaf: [4–12]<br>max_features: ['auto', 'sqrt','log2', None] | n_estimators: 344<br>learning_rate: 1.0<br>subsample: 1.0<br>max_depth: 1067<br>min_samples_split: 5<br>min_samples_leaf: 12<br>max_features: 'auto' |

**Table 6.** *Cont.*

| Classifier Name | Adjusted Parameters | Best GA-Based Configuration |
|---|---|---|
| LightGBM | boosting_type: ['gbdt', 'dart', 'goss', 'rf']num_leaves: [5–42]<br>max_depth: [10–1200] + None<br>learning_rate: $[1 \times 10^{-3}, 1 \times 10^{-2}, 1 \times 10^{-1}, 0.5, 1.]$<br>n_estimators: [100–1200]<br>min_child_samples: [100,500]<br>min_child_weight: $[1 \times 10^{-5}, 1 \times 10^{-3}, 1 \times 10^{-2}, 1 \times 10^{-1}, 1, 10, 100, 1000, 10000]$<br>subsample: sp_uniform(loc = 0.2, scale = 0.8)<br>colsample_bytree': sp_uniform(loc = 0.4, scale = 0.6)<br>reg_alpha: $[0, 10^{-1}, 1, 2, 5, 7, 10, 50, 100]$,<br>reg_lambda: $[0, 10^{-1}, 1, 5, 10, 20, 50, 100]$,<br>min_split_gain: 0.0,<br>subsample_for_bin: 200,000 | boosting_type: 'gbdt'<br>num_leaves: 13<br>max_depth: 15<br>learning_rate: 0.5<br>n_estimators: 500<br>min_child_samples: 399<br>min_child_weight: 0.1<br><br>subsample: 0.855<br>colsample_bytree: 0.9234<br><br>reg_alpha: 2<br>reg_lambda: 5<br>min_split_gain: 0.0,<br>subsample_for_bin: 200,000 |

**Table 7.** The accuracy of the optimized ensemble models for Dataset 1.

| Fold | GA–RF (%) | GA–AdaBoost (%) | GA–XGBoost (%) | GA–Bagging (%) | GA–GradientBoost (%) | GA–LightGBM (%) |
|---|---|---|---|---|---|---|
| 1 | 97.11 | 94.85 | 96.75 | 96.56 | 97.11 | 96.84 |
| 2 | 96.84 | 93.13 | 97.02 | 96.75 | 96.93 | 96.47 |
| 3 | 97.20 | 93.04 | 96.93 | 96.56 | 96.93 | 95.66 |
| 4 | 96.02 | 93.76 | 97.47 | 97.65 | 97.83 | 96.20 |
| 5 | 96.29 | 92.95 | 97.02 | 97.02 | 97.02 | 96.20 |
| 6 | 96.47 | 93.57 | 96.92 | 96.74 | 97.01 | 96.20 |
| 7 | 96.74 | 92.85 | 97.29 | 97.01 | 97.29 | 96.83 |
| 8 | 97.83 | 95.66 | 97.56 | 97.47 | 97.83 | 97.47 |
| 9 | 97.01 | 92.85 | 97.29 | 97.01 | 97.19 | 96.56 |
| 10 | 95.93 | 93.67 | 95.93 | 96.20 | 96.11 | 95.75 |
| **Average** | 96.74 | 93.63 | 97.01 | 96.90 | 97.13 | 96.42 |

To explore this further in Dataset 1, the confusion matrices of Random Forests, XG-Boost, Gradient Boost, and LightGBM are shown in Figures 5–8, respectively. Table 8 lists the results of the other performance measures.

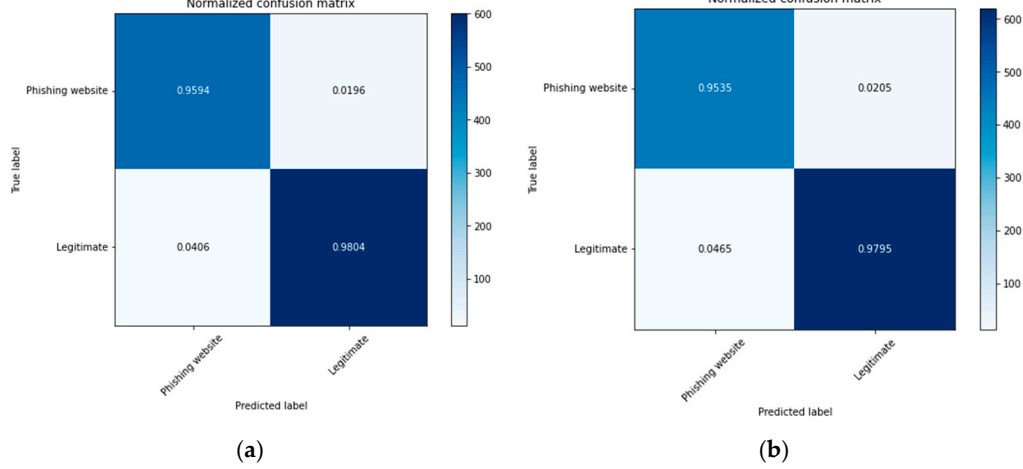

**Figure 5.** A normalized confusion matrix of Random Forests for phishing website and legitimate website classification: (**a**) with default parameters; (**b**) with optimized parameters for Dataset 1.

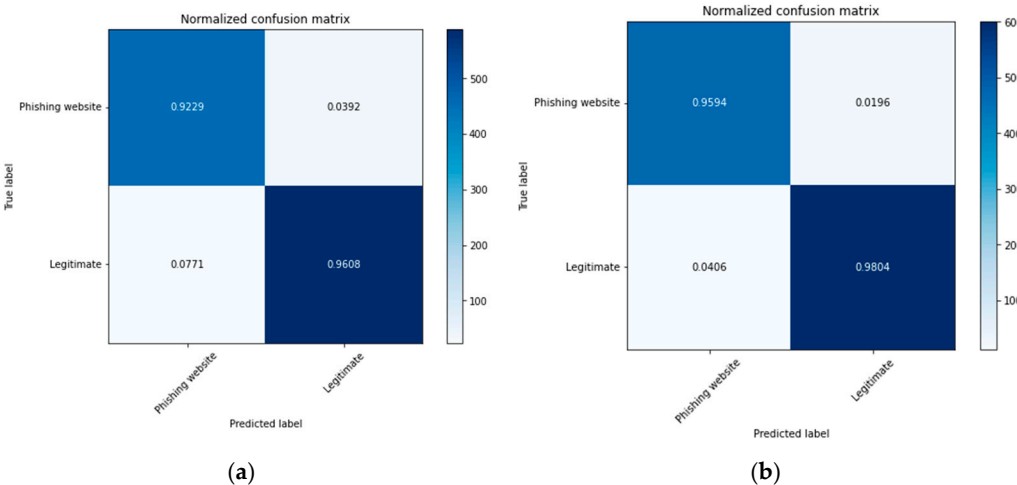

(**a**)                              (**b**)

**Figure 6.** A normalized confusion matrix of XGBoost for phishing website and legitimate website classification: (**a**) with default parameters; (**b**) with optimized parameters for Dataset 1.

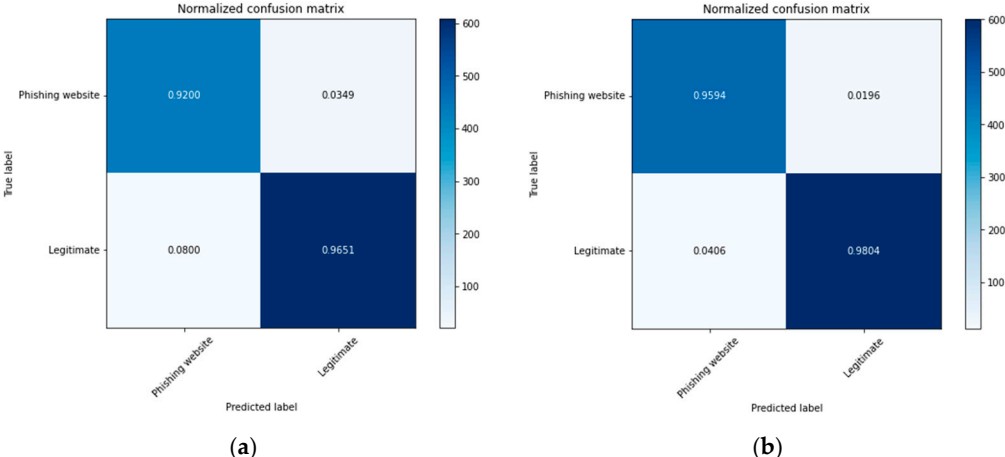

(**a**)                              (**b**)

**Figure 7.** A normalized confusion matrix of GradientBoost for phishing website and legitimate website classification: (**a**) with default parameters; (**b**) with optimized parameters for Dataset 1.

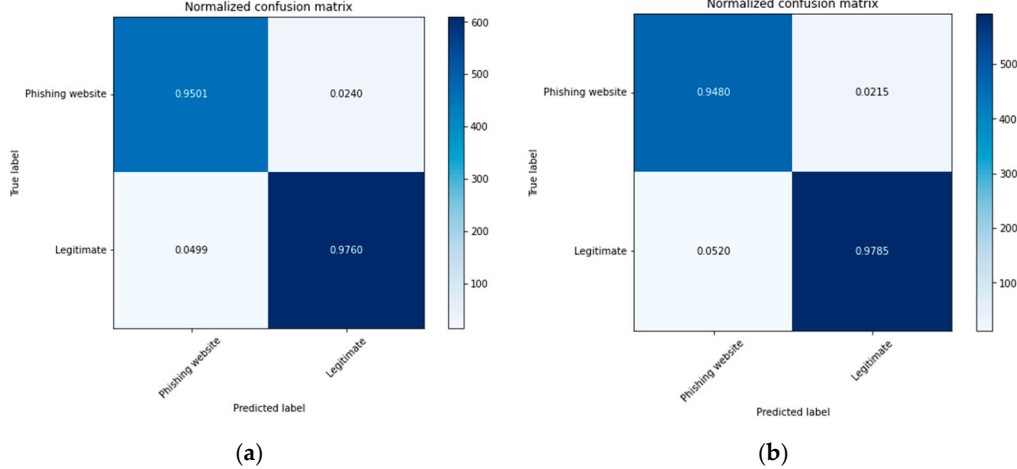

(**a**)                              (**b**)

**Figure 8.** A normalized confusion matrix of LightGBM for phishing website and legitimate website classification: (**a**) with default parameters; (**b**) with optimized parameters for Dataset 1.

In Figures 6b and 7b, we can note that the GA–XGBoost and GA–GradientBoost classifiers gained the most benefit from the optimization for Dataset 1. They correctly detected 95.94% of phishing website instances as "phishing website" class, which represents the TP measure, and incorrectly detected 4.06% of these instances as "legitimate" class, which represents the FP measure. In addition, they detected 98.4% of legitimate website instances as "legitimate" class, which represents the TN measure, and incorrectly detected 1.96% of these instances as "phishing website" class, which represents the FN measure. We can conclude that both classifiers (GA–XGBoost and GA–GradientBoost) achieved a high TP rate and a low FP rate.

**Table 8.** Results of performance evaluation measures when detecting phishing and legitimate classes for Dataset 1.

| Classifier | Class Name | Precision (%) | Recall (%) | F-Score (%) |
|---|---|---|---|---|
| GA–Random Forests | Phishing website | 96.40 | 94.10 | 95.10 |
| | Legitimate | 95.20 | 97.30 | 96.40 |
| | Weighted Average | 95.90 | 95.70 | 95.90 |
| GA–XGBoost | Phishing website | 97.50 | 95.80 | 96.50 |
| | Legitimate | 96.70 | 98.00 | 97.20 |
| | Weighted Average | 97.00 | 97.00 | 97.00 |
| GA–GradientBoost | Phishing website | 97.00 | 95.70 | 96.40 |
| | Legitimate | 96.80 | 97.50 | 97.10 |
| | Weighted Average | 96.90 | 96.80 | 96.80 |
| GA–LightGBM | Phishing website | 95.10 | 94.20 | 94.70 |
| | Legitimate | 95.50 | 96.30 | 95.80 |
| | Weighted Average | 95.30 | 95.30 | 95.30 |

After conducting the training phase for the ensemble classifiers on Dataset 1, the performances of these classifiers were ranked, and the three best models were: GA–GradientBoost, GA–XGBoost, and GA–Bagging. These models were used in the next step as base classifiers (base learner) of a stacking ensemble method. For Dataset 1, the classifiers that were used as meta-learners were Random Forests, GradientBoost, and Support Vector Machine (SVM).

The same experiments were conducted on Dataset 2 and Dataset 3. The performance of GA-based ensemble classifiers after optimization is shown in Tables 9 and 10. The results indicate that some classifiers (such as GA–Random Forests, GA–AdaBoost, and GA–XGBoost) show improvements in terms of accuracy, precision, recall, and F-score for Dataset 2, while all of the classifiers show improvements for Dataset 3 using all measures.

**Table 9.** Performance of GA-based ensemble classifiers for Dataset 2.

| Measure | GA–Random Forests (%) | GA–AdaBoost (%) | GA–XGBoost (%) | GA–Bagging (%) | GA–GradientBoost (%) | GA–LightGBM (%) |
|---|---|---|---|---|---|---|
| Accuracy | 98.39 | 97.21 | **98.57** | 97.51 | 98.50 | 98.32 |
| Precision | 98.46 | 97.15 | **98.50** | 97.24 | 98.31 | 98.10 |
| Recall | 98.13 | 97.28 | **98.64** | 97.89 | 98.54 | 98.56 |
| F-Score | 98.43 | 97.21 | **98.57** | 97.52 | 98.37 | 98.33 |

**Table 10.** Performance of GA-based ensemble classifiers for Dataset 3.

| Measure | GA–Random Forests (%) | GA–AdaBoost (%) | GA–XGBoost (%) | GA–Bagging (%) | GA–GradientBoost (%) | GA–LightGBM (%) |
|---|---|---|---|---|---|---|
| Accuracy | 96.44 | 94.06 | **97.35** | 96.96 | 97.27 | 97.21 |
| Precision | 94.63 | 90.91 | **96.20** | 95.3 | 95.68 | 96.13 |
| Recall | 95.08 | 92.02 | **96.14** | 95.96 | 96.30 | 95.81 |
| F-Score | 94.86 | 91.46 | **96.17** | 95.64 | 95.78 | 95.96 |

Table 11 shows the mean rank calculated for all classifiers for all three datasets. The results were obtained by 10-fold cross-validation before and after applying GA optimization.

**Table 11.** Models ranked by accuracy of classifier obtained by 10-fold cross-validation.

| ML Classifier | Dataset 1 | | | Dataset 2 | | | Dataset 3 | | |
|---|---|---|---|---|---|---|---|---|---|
| | Mean Rank | Mean | SD | Mean Rank | Mean | SD | Mean Rank | Mean | SD |
| RF | 3.2 | 0.970 | 0.00427 | 4.2 | 0.9837 | 0.00332 | 3.6 | 0.971 | 0.00229 |
| GA–RF | 4.4 | 0.967 | 0.00554 | 4.1 | 0.9839 | 0.00327 | 8 | 0.964 | 0.00269 |
| AdaB | 11.7 | 0.932 | 0.00549 | 11.5 | 0.9688 | 0.00477 | 12 | 0.936 | 0.00335 |
| GA–AdaB | 11.2 | 0.936 | 0.00889 | 10.2 | 0.9721 | 0.00448 | 11 | 0.941 | 0.00299 |
| XGB | 9.6 | 0.945 | 0.00491 | 7.4 | 0.9770 | 0.00508 | 9.5 | 0.9532 | 0.00339 |
| GA–XGB | 3.1 | 0.970 | 0.00447 | 2.5 | 0.9857 | 0.00310 | 1.5 | 0.974 | 0.00201 |
| Bagging | 5.3 | 0.967 | 0.00492 | 8.9 | 0.9751 | 0.00567 | 5.8 | 0.968 | 0.00214 |
| GA–Bagging | 4.3 | 0.969 | 0.00412 | 8.1 | 0.9751 | 0.00579 | 4.9 | 0.969 | 0.00243 |
| GB | 9.2 | 0.946 | 0.00578 | 7.8 | 0.9767 | 0.00492 | 9.5 | 0.954 | 0.00330 |
| GA–GB | 1.8 | 0.971 | 0.00464 | 2.9 | 0.9850 | 0.00293 | 2.2 | 0.973 | 0.00222 |
| LGB | 6.1 | 0.965 | 0.00561 | 1.9 | 0.9865 | 0.00307 | 6.7 | 0.967 | 0.00227 |
| GA–LGB | 6.7 | 0.964 | 0.00514 | 4.5 | 0.9832 | 0.00421 | 2.9 | 0.972 | 0.00235 |

The results show that most of the models with the highest mean accuracy values were produced when the GA was used. Among all of the selected classifiers, GA–XGB is a good choice for use as a base classifier for the stacking ensemble method.

Table 12 shows the testing results for the detection accuracy of the proposed model using 10-fold cross-validation for Dataset 1, Dataset 2, and Dataset 3.

**Table 12.** The accuracy of the optimized stacking ensemble method.

| Dataset | RF Level (%) | GB (%) | SVM (%) |
|---|---|---|---|
| Dataset 1 | 97.00 | 96.82 | 97.16 |
| Dataset 2 | 98.57 | 98.47 | 98.58 |
| Dataset 3 | 97.22 | 97.32 | 97.39 |

As shown in Table 12 above, the proposed optimized stacking ensemble model obtained good improvements in terms of phishing website detection accuracy for all datasets. The proposed optimized stacking ensemble obtained the best performance when the optimized ensemble classifiers (GA–GradientBoost, GA–XGBoost, and GA–Bagging) were used as base learners, and SVM was used as meta-learner. The achieved accuracy reached 97.16%, 98.58%, and 97.39% for Dataset 1, Dataset 2, and Dataset 3, respectively, which surpasses the other ensemble methods in the previous phase.

### 4.3. Statistical Analysis and Comparison with Previous Studies

Table 13 presents a comparison of the results obtained (using Dataset 1) with the preliminary settings, where the base classifiers were trained using the default settings of hyperparameters, and the improvements obtained after applying the GA and adjusting the hyperparameters of the classifiers. It also summarizes the mean accuracy and variance values of each classifier. The results also show that the mean of the GradientBoost classifier using GA optimization exceeded the means of all of the other classifiers, before and after applying the optimization.

**Table 13.** The average accuracy and variance values of all of the classifiers, before and after conducting GA optimization.

| Classifier Name | | Without Optimization | With GA Optimization |
|---|---|---|---|
| Random Forests | Avg. | **97.02%** | 96.74% |
| | Variance | 0.000 | 0.000 |
| AdaBoost | Avg. | 93.17% | **93.63%** |
| | Variance | 0.000 | 0.000 |
| XGBoost | Avg. | 94.45% | **97.01%** |
| | Variance | 0.000 | 0.000 |
| Bagging | Avg. | 96.73% | **96.90%** |
| | Variance | 0.000 | 0.000 |
| GradientBoost | Avg. | 94.61 | **97.13%** |
| | Variance | 0.000 | 0.000 |
| LightGBM | Avg. | **96.53%** | 96.42% |
| | Variance | 0.000 | 0.000 |

In addition to the basic statistical measures listed above, we measured the statistical significance of the results before and after applying optimization. Hence, the paired two samples were used for the mean t-test. The null hypothesis, h_0, for this comparison is that the mean accuracy values achieved before and after applying GA optimization to the classifiers are the same. The *p* values suggest that the null hypothesis can be rejected in four cases (out of six), which means that the improvement is significant in most of the cases (see Table 14).

**Table 14.** The reported *p* values for t-tests.

| Classifier Name | | *t*-Test Result | Conclusion |
|---|---|---|---|
| Random Forests | t-stat. | 1.466706885 | No significant |
| | *p*-value | 0.088 | improvement |
| AdaBoost | t-stat. | −2.100040666 | Significant |
| | *p*-value | 0.032556993 | improvement |
| XGBoost | t-stat. | −13.49130461 | Significant |
| | *p*-value | 0.000 | improvement |
| Bagging | t-stat. | −2.976672182 | Significant |
| | *p*-value | 0.008 | improvement |
| GradientBoost | t-stat. | −11.26647694 | Significant |
| | *p*-value | 0.000 | improvement |
| LightGBM | t-stat. | 0.971025 | No significant |
| | *p*-value | 0.178454 | improvement |

Similarly, the statistical analysis was conducted on the other datasets. It was found that the improvements obtained by AdaBoost, XGBoost, and GradientBoost with GA optimization were significant using Dataset 2, while for Dataset 3, the improvements obtained by all GA-based ensemble classifiers (except Random Forests) were significant.

In addition, the Friedman test results showed a significant difference in accuracy, of $(X^2 = 51.96, df = 9, p = 2.82 \times 10^{-7})$ for the first data set, and $(X^2 = 48.16, df = 9, p = 1.83 \times 10^{-5})$ and $(X^2 = 41.26, df = 9, p = 2.68 \times 10^{-5})$ for the second and third datasets, respectively. This indicates that it is safe to reject the null hypothesis when a model performed the same. In addition, we can conclude that at least one model has different performance values. Therefore, we conducted the Nemenyi post-hoc.

The comparative analysis of all of the models using their mean ranks was carried out. The calculated values of critical difference for the datasets were $CD = 4.9493$, $CD = 4.4094$, and $CD = 3.283$ for the first, second, and third datasets, respectively. Figures 9–11 show the critical difference diagrams where the models with statistically similar values of performance are connected to one another.

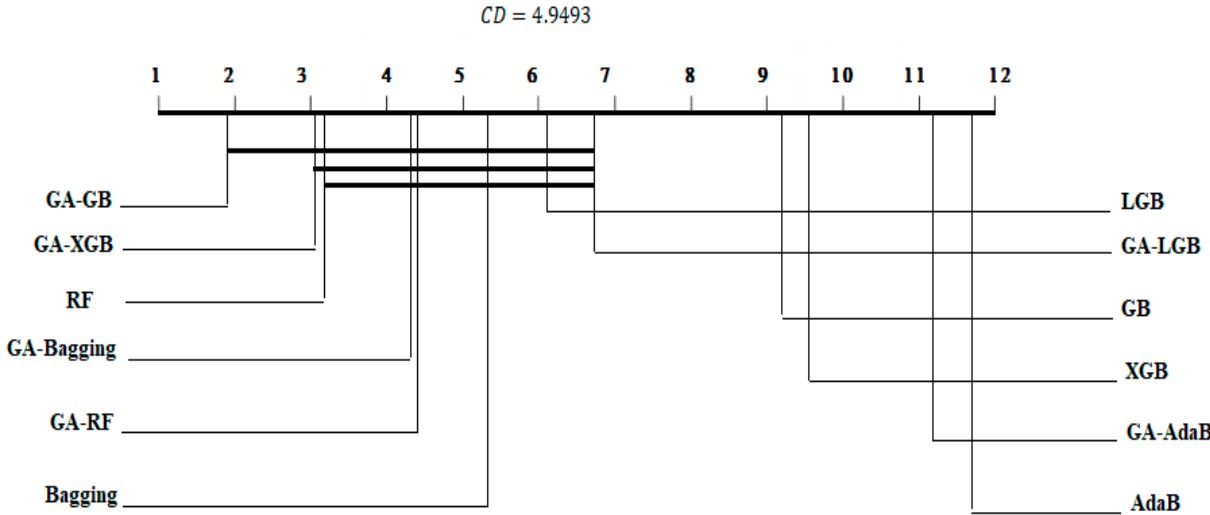

**Figure 9.** Critical difference diagram of Dataset 1 for the Nemenyi test.

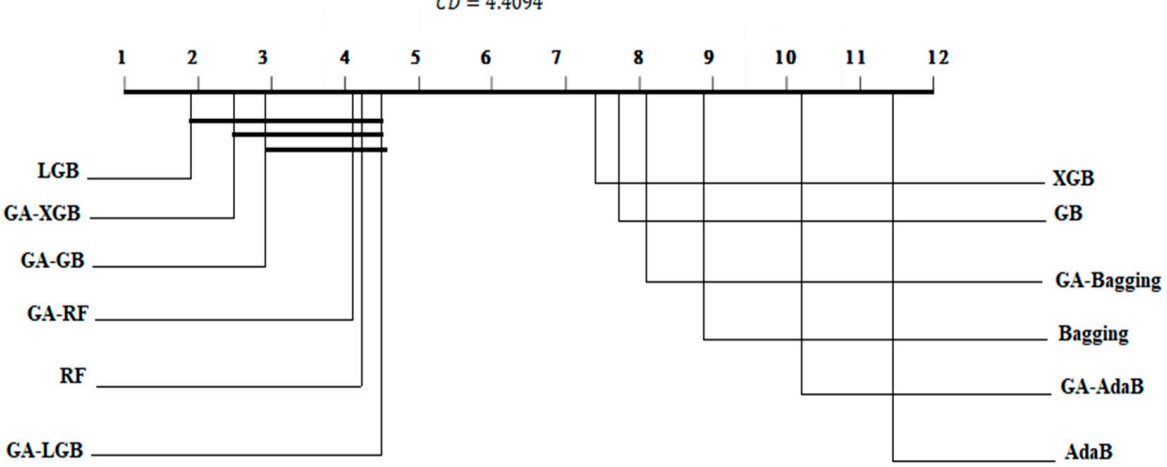

**Figure 10.** Critical difference diagram of Dataset 2 for the Nemenyi test.

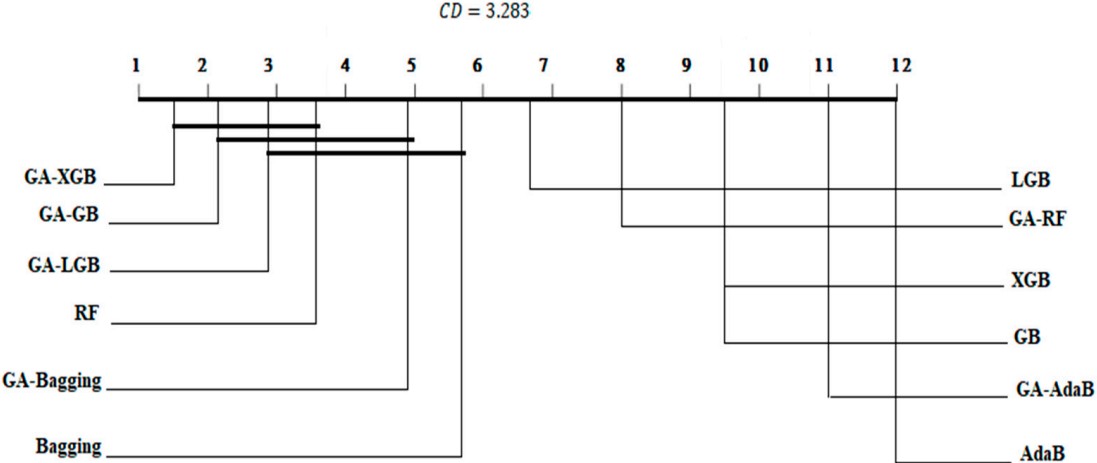

**Figure 11.** Critical difference diagram of Dataset 3 for the Nemenyi test.

Figure 9 shows the results of the statistical comparison of all of the models against one another by their mean ranks using Dataset 1 (higher ranks, such as 1.8 for GA–GB, correspond to higher values). Classifiers (only the three classifiers that have the highest values) that are not connected by a bold line of length equal to the CD have significantly different mean ranks (Confidence level of 95%).

Figure 10 shows the results of the statistical comparison of all of the models against one another by their mean ranks using Dataset 2 (higher ranks, such as 1.9 for LGB, correspond to higher values). Classifiers (only the three classifiers that have the highest values) that are not connected by a bold line of length equal to the CD have significantly different mean ranks (Confidence level of 95%).

Figure 11 shows the results of the statistical comparison of all of the models against one another by their mean ranks using Dataset 3 (higher ranks, such as 1.5 for GA–XGB, correspond to higher values). Classifiers (only the three classifiers that have the highest value) that are not connected by a bold line of length equal to the CD have significantly different mean ranks (Confidence level of 95%).

In addition, a comparison was conducted with the previous studies that used the same phishing websites (Dataset 1 and Dataset 2), which is presented in Table 15. As Dataset 3 was only recently prepared, it was not used in the previous studies. The evaluation metrics were accuracy, precision, and recall. The results show that the proposed optimized stacking ensemble method outperformed the other recent and related works [7,10] in using the accuracy and recall performance measures for Dataset 1, and outperformed [35] in using the accuracy, precision and recall measures for for Dataset 2.

**Table 15.** Comparison of the proposed method with the previous studies.

| Paper | Classifier | Dataset | Accuracy% | Precision % | Recall % |
|---|---|---|---|---|---|
| Ali and Ahmed [7] | GA–ANN | Dataset 1 | 88.77 | 85.81% | 93.34% |
| Ali and Malebary [10] | POS–RF | Dataset 1 | 96.83 | **98.76%** | 95.37% |
| This study | The stacking ensemble method | Dataset 1 | **97.16** | 96.86% | **96.83%** |
| Khan, Khan, and Hussain [35] | ANN after PCA | Dataset 2 | 97.13 | 96.48% | 98.03% |
| This study | The stacking ensemble method | Dataset 2 | **98.58** | **98.50%** | **98.74%** |

## 5. Conclusions

This paper has proposed an optimized stacking ensemble model for detecting phishing websites. In the optimisation method, a genetic algorithm, was used to find the optimized values for the parameters of several ensemble learning methods. The proposed model includes three phases: the training, ranking, and testing phases. In the training phase, several ensemble learning methods were trained without applying the optimization method (GA); these included Random Forests, AdaBoost, XGBoost, Bagging, GradientBoost, and LightGBM. These classifiers were then optimized using a GA that selects the optimal values of model parameters and improves their overall accuracy. In the ranking phase, the best three ensemble methods were selected and used as base classifiers for a stacking ensemble method. The stacking method also used three classifiers as meta-learners: RF, GB, and SVM. Finally, in the testing phase, new websites were collected and used as a testing dataset in order to predict the final class label of these websites (phishing or legitimate). The experimental results showed that the proposed optimized stacking ensemble method obtained superior performance compared to other machine-learning-based detection methods; the obtained accuracy reached 97.16%. A statistical analysis was conducted, which showed that the obtained improvements were statistically significant. In addition, the proposed methods were compared with recent studies that used the same phishing dataset, and it was reported that the proposed method surpassed those used in these studies. As phishing attacks are more dangerous in internet of things (IoT) environments—because IoT

devices are an easy medium for attackers, who can simply use the software in the thingbots for spreading spam emails without the user knowing—a light detection method will be proposed in future work to be applied to IoT environments. In addition, deep learning methods will be investigated in order to improve the detection rate of phishing websites, and more phishing datasets will be used.

**Author Contributions:** Conceptualization, Z.G.A.-M. and B.A.M.; methodology, M.A.-S. and F.S.; software, M.A.-S.; validation, M.A.-S., and T.A.-H., and A.A.; formal analysis, F.S.; investigation, T.A.-H., M.T.A. and T.S.A.; resources, F.S. and T.A.-H.; data curation, M.A.-S. and F.S.; writing—original draft preparation, M.A.-S., F.S. and B.A.M.; writing—review and editing, M.A.-S., F.S. and Z.G.A.-M.; visualization, F.S.; supervision, M.A.-S.; project administration, M.A.-S. and F.S.; funding acquisition, Z.G.A.-M., B.A.M., M.T.A., A.A and T.S.A.; All authors have read and agreed to the published version of the manuscript.

**Funding:** This research has been funded by the Scientific Research Deanship at the University of Ha'il, Saudi Arabia, through project number RG-20 023.

**Data Availability Statement:** Data are available in [31–33].

**Acknowledgments:** We would like to acknowledge the Scientific Research Deanship at the University of Ha'il, Saudi Arabia, for funding this research.

**Conflicts of Interest:** The authors declare no conflict of interest.

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
