# Peer review of "An Optimized Stacking Ensemble Model for Phishing Websites Detection"

_electronics, doi:10.3390/electronics10111285_

Round 1
Reviewer 1 Report
This paper proposes an optimized ensemble classification method for phishing website detection. Genetic Algorithm is utilized in the parameter finetuning procedure of several ensemble machine learning methods, including Random Forest, AdaBoost, XGBoost, Bagging, GradientBoost, and LightGBM. In the ranking phase, the optimized classifiers are ranked. Author select the best three models, which are GA-GradientBoost, GA-XGBoost and GA-Bagging. These selected models are then used by stacking method to integrate many machine learning methods. Experimental results shows that the proposed optimized stacking ensemble method obtained superior performance compared to other machine learning-based detection methods.
Suggestions:
- The author makes a statistical analysis to indicate that the obtained improvement by GA algorithm are statistically significant. But the results seem confusing. In Table 11, classifiers such as Random Forest obtain a lower accuracy with GA optimization. The purpose of t-test in Table 12 seems only to prove the change of accuracy when using GA algorithm. These result cannot demonstrate the benefit of GA algorithm convincingly.
- This paper just make a comparative study with other methods that using the same dataset. The proposed method can be tested on other datasets in order to obtain a more convincing result.
- Writing needs to be improved.
Author Response
Please find the response to your comments.
Regards
Tawfik

Reviewer 2 Report
The paper proposes an optimized ensemble classification method for phishing website detection. The method uses genetic programming to optimize the parameters of machine learning methods for classification. The results of experiments on a phishing website dataset are presented.
Comments:
- Line 28: “results showed significant improvements” – the improvement over previous methods is small. It is not supported by statistical analysis results.
- The connection of paper to IoT environments is rather artificial. The article just performs the recognition of malicious websites by their URL. The dataset is also not related to IoT.
- The novelty of the paper is minor. The paper does not propose a new model or method, but rather applies a combination of known methods for solving a specific problem. Clearly state the novelty and contribution of the paper in the field of security over the previous works at the end of the Introduction section.
- Section 2.1 title is misleading. There is no such thing as “IoT- based Phishing Detection Methods”, however, some of the discussed papers address the problem of recognizing phishing attacks in IoT.
- The discussion of previous works is weak. The selection of papers used for discussion seems to be ad hoc. Additionally, I suggest to discuss “A Stacked Deep Learning Approach for IoT Cyberattack Detection”, and “Identifying phishing attacks in communication networks using URL consistency features”. Summarize the limitations of previous works as a motivation for your study.
- Figure 1: The testing dataset is missing.
- Equation1 (1)-(6) are well known and could be omitted.
- Tables 2-5, 8, 10: there is no reason to present the results for each classification fold. Rather, present average and standard deviation or the 95% confidence limits and visualize using boxplots.
- For statistical comparison of ensemble classifiers, apply the post-hoc Nemenyi test and present the Critical Distance diagrams with the mean ranks of classifiers and Critical Distance (CD) value.
- Present a motivation for value selection in Table 6.
- The paired sample t-test assumes that the dependent variable should be approximately normally distributed. Did you check for the normality of distribution?
- Table 13 should be extended. There are many other works that used the same dataset. For example, “Comparative Evaluation of Techniques for Detection of Phishing URLs” achieved better performance on the same dataset.
Author Response

(The authors gave the same response as above.)

Reviewer 3 Report
This study proposes a stacking ensemble model to detect 317
phishing websites. As the optimization method a genetic algorithm is used to optimize the parameters of several ensemble learning methods. The manuscript is organized well and is easy to follow. However, the following comments are suggested for more improvement:
- In the section of related work, it's suggested to have a short review on the studies that focus on feature selection for phishing attacks. You may consider the work as below:
- M. Rajab, “An anti-phishing method based on feature analysis,” in
Proceedings of the 2nd International Conference on Machine Learning and Soft Computing. ACM, 2018, pp. 133–139. - M. Babagoli, M. P. Aghababa, and V. Solouk, “Heuristic nonlinear
regression strategy for detecting phishing websites,” Soft Computing,
pp. 1–13, 2018. - Zabihimayvan, Mahdieh, and Derek Doran. "Fuzzy rough set feature selection to enhance phishing attack detection." 2019 IEEE International Conference on Fuzzy Systems (FUZZ-IEEE). IEEE, 2019.
- M. Rajab, “An anti-phishing method based on feature analysis,” in
- Regarding the data used in this work, it is strongly suggested to try other online datasets such as:
- https://data.mendeley.com/datasets/h3cgnj8hft/1
- https://data.mendeley.com/datasets/72ptz43s9v/1
It helps to better indicate the performance of your work on a more general data.
- In tables 2-5, please highlight or bold the numbers that you discuss in the text.
Author Response

(The authors gave the same response as above.)

Round 2
Reviewer 2 Report
Unfortunately, the authors have ignored a majority of my comments in the previous review. I urge the authors to address the comments of the reviewers more seriously.
- The novelty of the presented approach has no been defined satisfactorily. All the methods mentioned by the authors (genetic algorithm (GA), Random Forest, AdaBoost, XGBoost, Bagging, GradientBoost, and LightGBM) are well known and have been used many times before.
- The discussion of related works is still weak. The authors added only one more work. It is suggested to discuss more related papers, including “Detection of malicious urls on twitter”, “Comparative evaluation of techniques for detection of phishing URLs”, and other recent works.
- Present a standard deviation or the 95% confidence limits of values presented in Table 7-11 and visualize using boxplots.
- For the statistical comparison of ensemble classifiers, apply the post-hoc Nemenyi test and present the Critical Distance diagrams with the mean ranks of classifiers and Critical Distance (CD) value.
- The authors did not compare their results with the results of other authors on the same dataset. I urge the authors to present such a comparison.
Author Response
Response to Reviewers
Dear Ms. Miroslava Bunic
Assistant Editor
Electronics
Thank you very much for your kind efforts on reviewing our paper and thank you very much for the kind reviewer for the useful feedback and comments that helped us to improve the quality of the paper.
Kindly find the below response to all comments given by Reviewers 1.
We are looking for your kind feedback.
Best regards,
Dr. Tawfik Al-Hadhrami,
CA
Reviewer 1:
Comment 1:
The novelty of the presented approach has no been defined satisfactorily. All the methods mentioned by the authors (genetic algorithm (GA), Random Forest, AdaBoost, XGBoost, Bagging, GradientBoost, and LightGBM) are well known and have been used many times before.
Response 1:
Thank you very much for your kind feedback that truly helped us to improve the quality of this paper. The proposed method could be considered as an approach for detecting the phishing website that is done through several steps, starting by extracting the websites features, apply GA optimization on several ensemble methods that were ranked and the best performing methods were used in the next step: staking ensemble as base classifiers. This method showed a great improvements in terms of accuracy, precision and recall. Also, it shows outperforming performance when comparing to recent methods that were applied on the same datasets.
Comment 2:
The discussion of related works is still weak. The authors added only one more work. It is suggested to discuss more related papers, including “Detection of malicious urls on twitter”, “Comparative evaluation of techniques for detection of phishing URLs”, and other recent works.
Response 2:
Thank you very much for this important comment. We have enriched the discussion with adding the suggested studies:
[17]. Alotaibi, B., & Alotaibi, M. (2020). A Stacked Deep Learning Approach for IoT Cyberattack Detection. Journal of Sensors, 2020.
[26]. Azeez, N. A., Salaudeen, B. B., Misra, S., Damaševičius, R., & Maskeliūnas, R. (2020). Identifying phishing attacks in communication networks using URL consistency features. International Journal of Electronic Security and Digital Forensics, 12(2), 200-213.
[27]. Azeez, N. A., Atiku, O., Misra, S., Adewumi, A., Ahuja, R., & Damasevicius, R. (2020). Detection of Malicious URLs on Twitter. In Advances in Electrical and Computer Technologies (pp. 309-318). Springer, Singapore.
[28]. Osho, O., Oluyomi, A., Misra, S., Ahuja, R., Damasevicius, R., & Maskeliunas, R. (2019, November). Comparative Evaluation of Techniques for Detection of Phishing URLs. In International Conference on Applied Informatics (pp. 385-394). Springer, Cham.
[35]. Khan, S. A., Khan, W., & Hussain, A. (2020, October). Phishing Attacks and Websites Classification Using Machine Learning and Multiple Datasets (A Comparative Analysis). In International Conference on Intelligent Computing (pp. 301-313). Springer, Cham.
Comment 3:
Present a standard deviation or the 95% confidence limits of values presented in Table 7-11 and visualize using boxplots.
Response 3:
Thank you very much for your useful suggestion. We have included the standard deviation for the accuracies of all methods using all datasets in Table 11.
Comment 4:
For the statistical comparison of ensemble classifiers, apply the post-hoc Nemenyi test and present the Critical Distance diagrams with the mean ranks of classifiers and Critical Distance (CD) value.
Response 4:
Thank you very much for this useful comment. The the post-hoc Nemenyi test has been applied, as shown in Section 4.3 and Figures 9, 10, &11.
Comment 5:
The authors did not compare their results with the results of other authors on the same dataset. I urge the authors to present such a comparison.
Response 5:
Thank you very much for this important suggestion. We have compared the proposed method on the recent methods applied on the same Datasets (1 & 2). For the Dataset 3, as it has been prepared recently, it has not been applied yet in the previous studies. Please refer to Table 15 in Section 4.3.

Round 3
Reviewer 2 Report
The manuscript was well revised. I have only a few minor comments.
- Present real dataset names in the abstract and elsewhere rather than invented. [31] is Phishing Websites Data Set, etc.
- Tables 2-4, 7, 10, 12: the units of measurement (%) are missing. Tables 8 and 13 use scaled values rather than percentages. Why?
Author Response
Response to Reviewers
Dear Ms. Miroslava Bunic
Assistant Editor
Electronics
Thank you very much for your kind efforts on reviewing our paper and thank you very much for the kind reviewer for the useful feedback and comments that helped us to improve the quality of the paper.
Kindly find the below response to all comments given by Reviewers 2.
We are looking for your kind feedback.
Best regards,
Dr. Tawfik Al-Hadhrami,
CA
Reviewer 2:
Comment 1:
Present real dataset names in the abstract and elsewhere rather than invented. [31] is Phishing Websites Data Set, etc.
Response 1:
Done. The real names of datasets have been added in Abstract and Dataset section.
Comment 2:
Tables 2-4, 7, 10, 12: the units of measurement (%) are missing. Tables 8 and 13 use scaled values rather than percentages. Why?
Response 2:
The measurement (%) has been added to the header of all tables. The values in Tables 8 and 13 have been revised and the measurement (%) was used.
